# Single-cell analysis identifies genes facilitating rhizobium infection in *Lotus japonicus*

Manuel Frank [1,5], Lavinia Ioana Fechete [1,5], Francesca Tedeschi[1], Marcin Nadzieja[1], Malita Malou Malekzadeh Nørgaard[1], Jesus Montiel [1,2], Kasper Røjkjær Andersen [1], Mikkel H. Schierup [3], Dugald Reid [1,4] ✉ & Stig Uggerhøj Andersen [1] ✉

Legume-rhizobium signaling during establishment of symbiotic nitrogen fixation restricts rhizobium colonization to specific cells. A limited number of root hair cells allow infection threads to form, and only a fraction of the epidermal infection threads progress to cortical layers to establish functional nodules. Here we use single-cell analysis to define the epidermal and cortical cell populations that respond to and facilitate rhizobium infection. We then identify high-confidence nodulation gene candidates based on their specific expression in these populations, pinpointing genes stably associated with infection across genotypes and time points. We show that one of these, which we name *SYMRKL1*, encodes a protein with an ectodomain predicted to be nearly identical to that of SYMRK and is required for normal infection thread formation. Our work disentangles cellular processes and transcriptional modules that were previously confounded due to lack of cellular resolution, providing a more detailed understanding of symbiotic interactions.

Plants require nutrients in order to grow and develop. One of the most important limiting nutrients is nitrogen, which is taken up by plants from the soil where it is mostly present as $NO_3^-$ and to a lesser extent as $NH_4^+$[1]. Legumes can grow independently of soil nitrogen by forming root nodules that host symbiotic, nitrogen-fixing rhizobia[2,3]. Nodule formation requires successful intracellular infection of the plant by the rhizobial symbionts. At the onset of the infection process in *Lotus japonicus* (Lotus), rhizobia and the host plant communicate intensively, leading to the curling of some root hairs (RHs) and subsequent infection pocket and infection thread (IT) formation[4,5]. After IT establishment in root hairs, cortical ITs are formed, linking the infected root hairs with the developing root nodule. The perception of rhizobia in Lotus root hairs triggers the formation of nodules that lose their meristematic activity over time (determinate nodules), while other

legumes like *Medicago truncatula* (Medicago) form indeterminate nodules that have a persistent meristem[6]. Determinate nodules are formed by dividing cortical cells[7] and indeterminate nodules originate from pericycle cells[8]. After nodule establishment and subsequent infection through cortical ITs, bacteria are released into bacteroids, ultimately enabling nitrogen fixation in mature nodules.

At the molecular level, symbiotic signaling is initiated by plant flavonoids that induce synthesis of rhizobial Nod Factors, which are in turn perceived by the receptor kinases NOD FACTOR RECEPTOR1 (NFR1) and NFR5[9–11]. Other receptor kinases involved in the early signaling events are RHIZOBIAL INFECTION RECEPTOR-LIKE KINASE1 (RINRK)[12], the bacterial exopolysaccharide-perceiving EXOPOLYSACCHARIDE RECEPTOR3 (EPR3)[13–15] and SYMBIOSIS RECEPTOR-LIKE KINASE (SYMRK)[16–18]. Structurally, the latter consists of an ectodomain

[1]Department of Molecular Biology and Genetics, Aarhus University, Universitetsbyen 81, DK-8000 Aarhus C, Denmark. [2]Center for Genomic Sciences, National Autonomous University of Mexico, Cuernavaca, Mexico. [3]Bioinformatics Research Centre, Aarhus University, Universitetsbyen 81, DK-8000 Aarhus C, Denmark. [4]Department of Animal, Plant and Soil Sciences, School of Agriculture, Biomedicine and Environment, La Trobe University, Melbourne, Australia. [5]These authors contributed equally: Manuel Frank, Lavinia Ioana Fechete. ✉e-mail: dugald.reid@latrobe.edu.au; sua@mbg.au.dk

harboring a malectin-like domain (MLD), a GDPC motif, three leucine-rich repeats and an intracellular kinase-domain[19]. The MLD domain has been shown to positively regulate protein stability and localization to the plasma membrane, yet negatively affects interaction with NFR5[20,21].

During the subsequent IT formation and growth, actin filaments rearrange[22,23] and plant cell walls are modified, which requires NODULATION PECTATE LYASE1 (NPL1)[24]. Moreover, plant hormones including auxin, cytokinin and ethylene are known to regulate IT formation and nodule organogenesis[6]. A number of transcriptional regulators are activated during infection and organogenesis, including ERF REQUIRED FOR NODULATION1 (ERN1)[25,26], CYCLOPS[27], NODULE INCEPTION (NIN)[28,29], NODULE SIGNALLING PROTEIN1 (NSP1) and NSP2[30] and NUCLEAR TRANSCRIPTION FACTOR YA1 (NF-YA1)[31]. Their loss of function results in impaired nodule initiation and infection. Loss-of-function *cyclops* mutants abort IT formation after the establishment of infection pockets and nodule organogenesis before forming a mature nodule, with infection failing to progress to the cortex[27]. Upon rhizobial infection, CYCLOPS regulates the expression of *ERN1*[32], and ectopic expression of *ERN1* rescues the *cyclops* infection phenotype[33].

Insight into the transcriptional responses of IT and nodule-forming cells is crucial to identify key regulators of IT establishment and nodule organogenesis and to fully understand the underlying mechanisms. However, the frequency of these events is relatively low within a tissue, and the transcriptional signatures of the few cells involved cannot be resolved from heterogeneous cell populations in classical approaches like bulk RNA-seq. To reduce this dilution effect, laser-capture microdissection[34,35], tissue enrichment approaches[36–38] and dissection of precise developmental zones[39] have been conducted. More recently, sequencing of single cells or nuclei has been used to study rhizobium infection and indeterminate nodule differentiation trajectories in Medicago[40,41]. The single-cell resolution and relatively large cell/nuclei counts afforded by 10x Genomics Chromium technology allowed the detection of a pervasive early response to rhizobium infection two days post inoculation (dpi) across root tissues and identification of a large number of infection-responsive genes, including differential responses of known nodulation genes across tissues[41]. In addition, a limited number of cells derived from determinate Lotus nodules were studied using a Smart-Seq2 protocol[42,43].

Here, we performed protoplast-based single-cell RNA-seq of Lotus wild-type seedlings ten dpi as well as wild-type and *cyclops* seedlings five dpi with *M. loti*. We focused the data analysis on the identification of carefully defined populations of cells representing specific stages of the infection and nodulation process in order to identify high-confidence candidate nodulation genes specifically expressed in each target population. *SYMRK-LIKE1* (*SYMRKL1*) represents a prominent example of cell population-specific expression, and we show that it is a regulator of rhizobium infection.

## Results

### The transcriptome of rhizobium-infected Lotus roots

To classify cells according to tissue type and determine cellular responses to rhizobium infection, we carried out single-cell RNA sequencing of protoplasts from mock-treated and rhizobium-inoculated Lotus roots ten dpi with two biological replicates per condition and about 150 whole roots per biological replicate. We characterized a total of 25,024 cells after filtering, 13,241 cells from control samples and 11,783 cells from inoculated samples (**Source Data "10 dpi Cells_Cell type"**), with a median of 2,859.5 unique molecular identifiers and ~1,500 transcripts per cell after filtering (**Source Data "seq1"**). We assessed the similarity of the samples by correlating the gene counts in the RNA assay after normalization. Control replicates had a Pearson correlation factor of 0.99 and inoculated replicates of 0.98 (**Source Data "Correlation 10dpi"**). The Pearson correlation factor between the control and the inoculated samples was in the 0.88

to 0.90 range. The samples were then integrated and clustered using Seurat[44], yielding 32 clusters (Fig. 1a, b, **Source Data "10dpi_CM"**). We determined cellular identities of individual clusters using marker gene information from *Lotus Base*[45], homologous markers from Arabidopsis and promoter-reporter lines[46] (Fig. 1a, b, **Source Data "10dpi_CM"**, Supplementary Fig. 1–3). All known Lotus root tissues were identified in both inoculated and uninoculated samples, indicating that the protoplasting had been effective even for deeper root tissues and vasculature. We found a substantial transcriptional response to rhizobial infection for most tissue types and subclusters within tissues, using the MAST algorithm with an adjusted *p* value ≤ 0.05 and a logFC ≥ |0.25|. The least responsive tissues were phloem, especially cluster 30, quiescent center cells (cluster 28) and xylem (cluster 29) (Fig. 1c, **Source Data "10dpi_DE_Genes"**). We observed that clusters responsive to inoculation, including clusters 4, 14 and 22, were over-represented in treated samples, while unresponsive clusters, e.g. clusters 29 (xylem) and 30 (phloem), were overrepresented in control samples (**Source Data "10 dpi Cells_cluster", "10 dpi Cells_Cell type"**, Supplementary Fig. 4a). As protoplasting can cause a stress-associated transcriptional response, we compared the transcriptomes of whole roots and protoplasts using bulk RNA-sequencing. We identified 655 genes that were induced by protoplasting and these were marked in the gene lists as "protoplast-induced" (**Source Data**).

### Identification of infected, nodule and bacteroid-containing cells

Having mapped cells to tissue types, we next focused on identifying subsets of cells specifically responding to rhizobial infection. The target populations were infected cells, harboring rhizobia within infection threads, nodule cells, which are not infected but form part of the nodule structure, and bacteroid cells, where nitrogen fixation takes place (Fig. 2a). To identify the infected cells, we reclustered the data using only the rhizobium-inoculated samples in order to emphasize the effect of the infection transcriptional response on the clustering (Supplementary Fig. 5b). Examining the expression patterns of the infection-related nodulation gene *NPL*[24] (Supplementary Fig. 3a) and other infection-related genes (**Source Data "VAL"**), we identified one cluster representing infected cells (Supplementary Fig. 5b). We could then highlight their positions in the original UMAP comprising all samples and identify infected cells in both root hairs (cluster 9) and cortex (cluster 8) (Figs. 1a and 2b). For the nodule cells, we found that cluster 14 showed specific expression of the nodule marker gene *CARBONIC ANHYDRASE*[47] (*ßCA1*, Supplementary Fig. 3b) and defined 98 *ßCA1*-expression cells in cluster 14 as nodule cells (Fig. 2c). Finally, we identified 21 cells from cluster 8 as bacteroid cells based on their expression of leghemoglobin genes *LB1*, *LB2* or *LB3*[48] (Fig. 2d and Supplementary Fig. 3c).

### Identification of candidate nodulation genes

To discover genes likely to be involved in the nodulation process, we selected candidates with expression patterns highly specific to each of the three populations by requiring that they were confidently identified as markers for the target population and be expressed in less than 2% of all other cells (**Source Data "10_INF", "10_NOD" and "10_BAC"**). We identified 592 genes matching these criteria, most of which were specific to a single population (Fig. 2e). There were also substantial overlaps, especially between infected and bacteroid cell candidates, and 21 out of 47 experimentally validated nodulation genes were included among the candidates (Fig. 2e and **Source Data**). The other 26 genes were found in these cell populations as well, but were expressed in more than 2% of all other cells. *NF-YA1* and *NIN* were identified as candidates for all three populations (Fig. 2e and **Source Data**).

To validate our approach, we used an independent set of root hair bulk RNA-seq data[38]. We reclustered the root hair cells and used Scissor[49] to identify a distinct subpopulation of 36 RH cells responding

to rhizobium inoculation (Fig. 2f and Supplementary Fig. 6). We compared the population of cells to the remaining RH cells and generated a marker gene list containing 67 genes (**Source Data "Scissor +"**). The marker gene list for this subcluster included well-known infection-related genes like *NPL*, *NF-YA1* and *NIN* and, as exemplified by *NPL*, were specific for this subpopulation (Fig. 2g and Supplementary Fig. 6c). We intersected the 149 infected cell marker genes and the 67 Scissor+ marker genes (Fig. 2h), identifying a large overlap of 50 genes.

## Infection responses in root hairs and cortical cells overlap but have distinct components

Successful nodulation requires progression of infection threads from epidermal to cortical cells. Whether the infection mechanism is conserved across these two cell types or cell-type specific is not well understood. To investigate this, we split the 96 infected cells by tissue, identifying 27 infected root hair and 69 infected cortical cells (Fig. 3a). These 27 infected RH cells were part of the 36 RH cells that were called

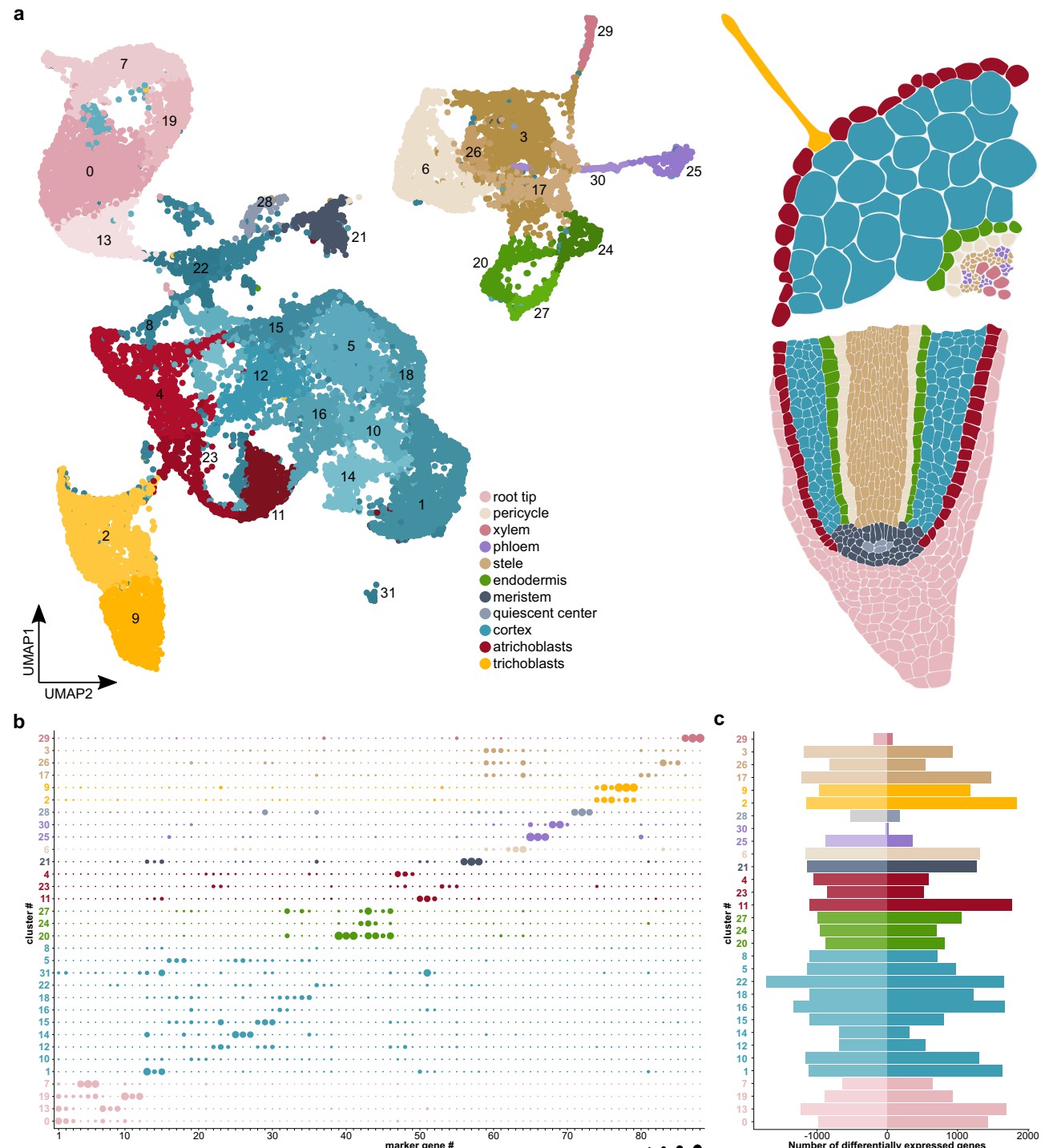

**Fig. 1 | A cellular atlas of rhizobium infection in Lotus roots. a** UMAP of control and *M. loti* inoculated root cells ten dpi showing clusters for known root cell types. **b** Dotplot depicting a selection of marker genes specific to the identified clusters. Confirmation of clusters by expression of marker gene reporter constructs in Lotus roots is depicted in Supplementary figure 1. **c)** Down- and Upregulation of genes in the clusters in response to rhizobial infection compared to wild-type. The background color of cluster numbers indicates the respective tissue identity. Lists of differentially expressed genes and marker genes can be found in **Source Data**.

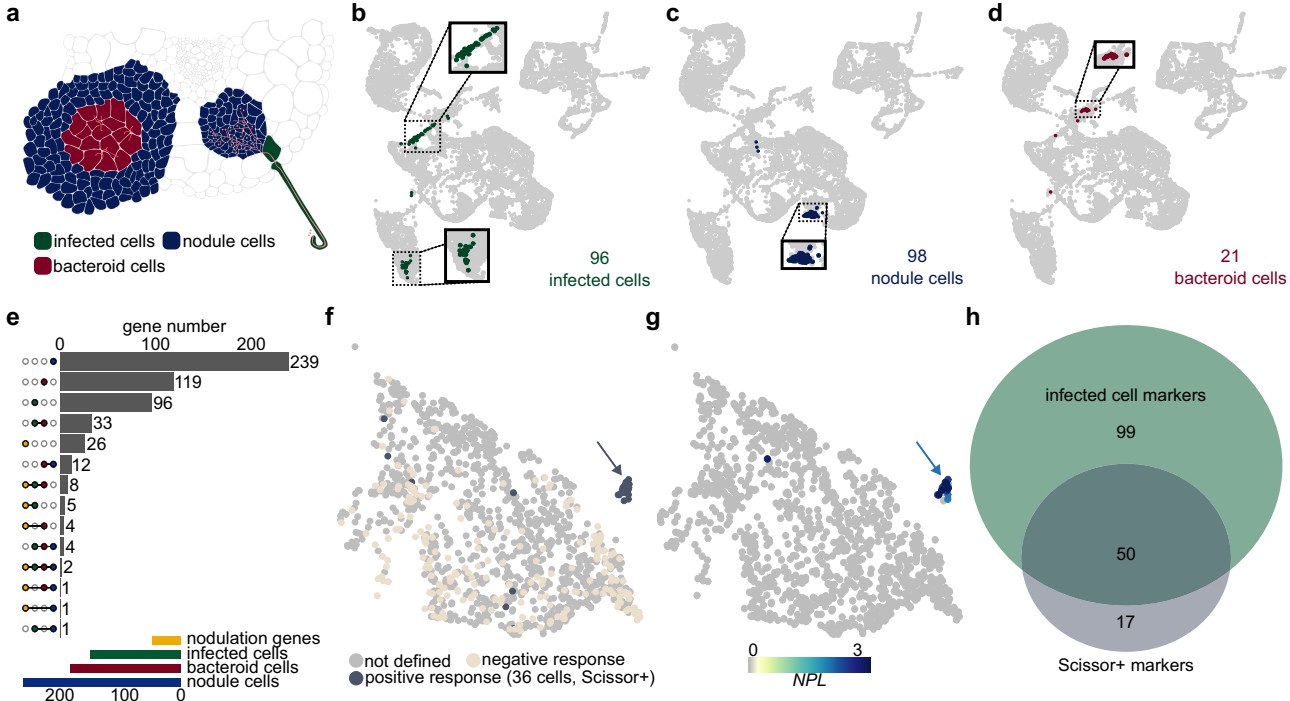

**Fig. 2 | Infected, nodule and bacteroid-containing cells in Lotus roots 10 dpi.**
**a** Illustration of infected cells (dark green), nodule cells (dark blue) and bacteroid cells (dark red) in a root cross-section. **b–d** UMAP of infected, nodule and bacteroid cells ten dpi. **e** Upset plot depicting the number of high confidence marker genes of infected, nodule and bacteroid-containing cells compared to a gene list of verified nodulation-related genes (**Source Data "VAL", "10_INF", "10_BAC", "10_NOD"**). **f** Identification of infected root hair cells by Scissor using recently published root hair bulk RNA-seq data[38]. Positively (Scissor + , dark grey) and negatively (sand-

colored) responding RH cells from *M.loti* inoculated samples are depicted, respectively. The cell population with the highest proportion of positively responding cells is indicated by a gray arrow. **g** Normalized expression of the Scissor+ cell marker gene *NPL*. **h** Venn diagram of infected cell and Scissor+ marker genes. RH subclusters, Scissor output and *NPL* expression of control and *M. loti* samples can be found in Supplementary Figure 4. The cell population with the highest proportion of *NPL* expressing cells is indicated by a blue arrow.

---

by Scissor as positively responding to rhizobia. We then selected genes showing strongly enriched expression in each population, identifying a root hair transcriptional module containing 18 genes and a larger cortex module comprising 81 genes (Fig. 3b, **Source Data "INF_RH" and "INF_C"**). In addition, 46 genes were equally expressed in both cell populations, constituting a mixed module (Fig. 3b, **Source Data "INF_RH_C"**). Exemplifying the three modules, *NPL* represents the common module (Fig. 3c, Supplementary Fig. 7a), *Lotja-Gi2g1v0018600* encoding an O-METHYLESTERASE (OMT) was enriched in infected cortical cells (Fig. 3d and Supplementary Fig. 7b) and *ISOPENTENYLTRANSFERASE4* (*IPT4*), encoding a cytokinin biosynthesis enzyme, was enriched in infected RH cells (Fig. 3e, Supplementary Fig. 7c).

### *Cyclops* mutants contain a unique population of responsive root hair cells

At 10 dpi, most infection threads are already fully elongated within root hairs, either arresting at that point or continuing to proliferate in cortical cells. To further differentiate between root hair and cortical programs, and to get a better understanding of the genes required for early infection events, we carried out single-cell RNA-sequencing using 150 susceptible zones per biological replicate at 5 dpi with water (control) or R7A. Within this experiment, we included the *cyclops* mutant and the wild type. We prepared two biological replicates per condition and genotype for a total of 8 samples. The *cyclops* mutant is characterized by abortion of infection thread formation after establishment of an infection pocket, and forms nodule primordia that fail to become infected[27]. We detected 32,180 high-quality cells of which about 11,000 originated from *cyclops* and 21,000 from wild-type samples (**Source Data "5 dpi Cells_Cell type"**). A median of 1865.5

unique molecular identifiers and ~1200 transcripts were detected per cell (**Source Data "seq2"**). Similarly to the 10 dpi samples, we were able to identify all known root tissues and the marker genes from the 10 dpi dataset were also specifically expressed in the 5 dpi dataset (Fig. 4a, Supplementary Fig. 1–3).

In the wild type 5 dpi samples, we did not observe expression of *ßCA*, indicating that mature, uninfected nodule cells were absent at this early time point. Likewise, we did not detect bacteroid cells displaying leghemoglobin expression, consistent with the absence of pink, nitrogen-fixing nodules. At 5 dpi we harvested the susceptible zone to enrich for infection events and nodule primordia. We selected infected and nodule primordia cells in the wild-type samples based on the expression of the well-described marker gene *NF-YA1* (Fig. 4b and Supplementary Fig. 8a), identifying 14 infected root hair and 330 nodule primordium cortical cells (Fig. 4b). The 5 dpi wild type infected/primordia cells showed specific expression of many of the same genes characteristic of 10 dpi infected cells, indicating stability across time for the infection transcriptional module and validating 69 of the 10 dpi candidate genes (Fig. 4c, **Source Data "5_NF-YA1"**).

We also found a number of *NF-YA1-expressing* root hair cells in the *cyclops* samples (Supplementary Fig. 8a), indicating that infection pocket formation is sufficient to induce *NF-YA1* expression. To determine what fraction of the infection transcriptional program was activated in the responsive *cyclops* root hairs, we re-clustered wild-type and *cyclops* root hair cells, identifying two rhizobium-responsive subclusters. One was enriched in the wild-type inoculated samples (Fig. 4d, Supplementary fig. 9a, b, subcluster 6) and the other in *cyclops* inoculated samples (Fig. 4d, Supplementary Fig. 9a, b, subcluster 5). A total of 42 genes enriched in the responsive *cyclops* root hair cells overlapped with genes specifically expressed in 10 dpi infected cells or

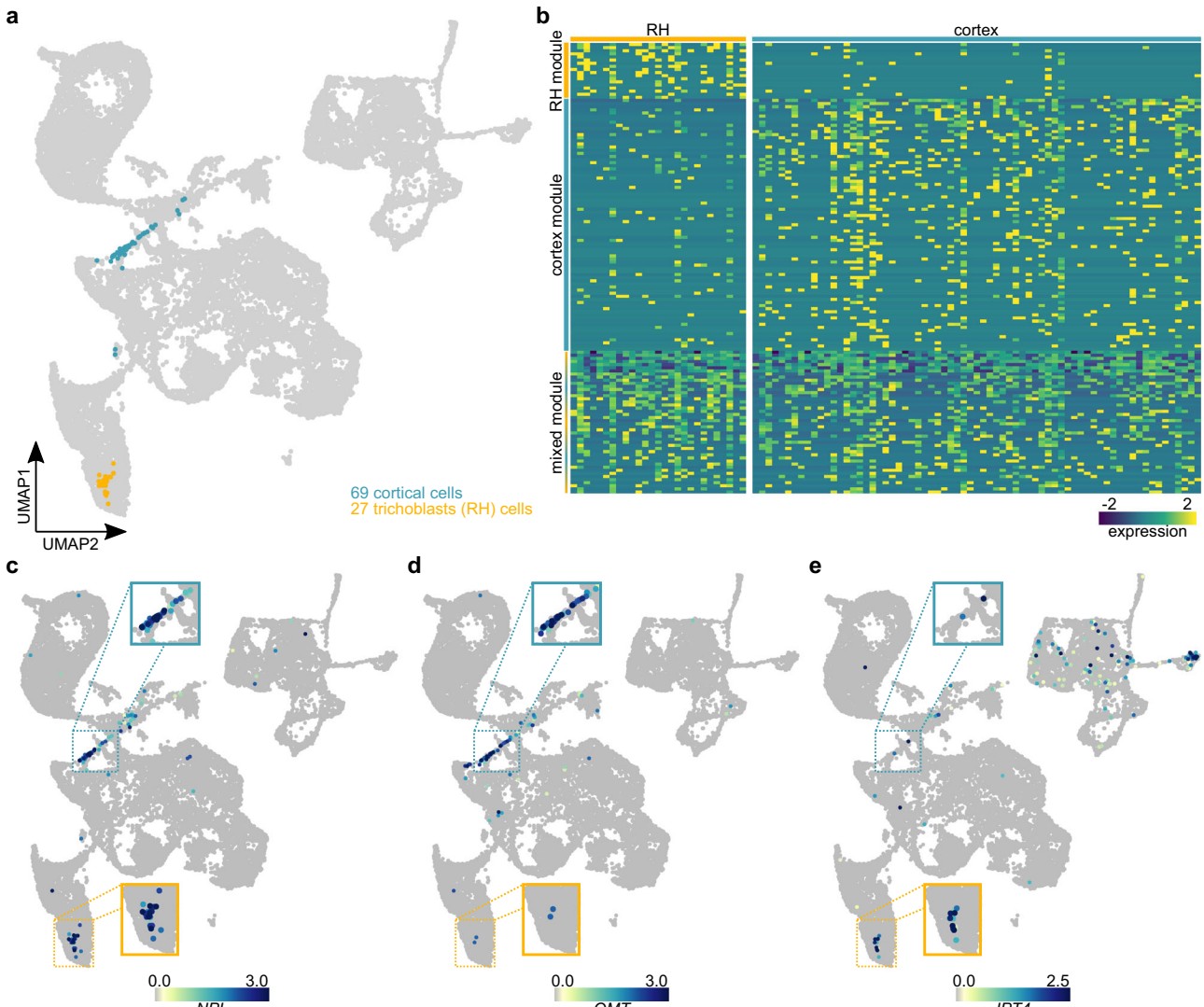

**Fig. 3 | Root hair- and cortex-IT cells share common and separate gene sets. a** Illustration of supposedly infected cortical (turquoise) and root hair (RH, yellow) cells of the ten dpi data set from *M. loti* treated plants. **b** Heatmap showing the expression of the marker genes for the RH module (yellow), cortex module (turquoise) and mixed module (yellow-turquoise). Normalized expression of the mixed module marker gene *NPL* **c**), the cortex module marker gene *OMT* **d**) and the RH module marker gene *IPT4* **e**) ten dpi. Areas containing infected cell populations are highlighted and cell populations belonging to either cortex or root hairs are marked with turquoise or yellow borders, respectively. Gene lists for the three modules can be found in **Source Data**.

wild type 5 dpi infected/primordia cells, indicating activation of a substantial part of the infection transcriptional program in the responding *cyclops* root hairs (Fig. 4c, **Source Data "5_Cyclops_RH5"**). The relatively few responsive wild-type root hair cells allowed the identification of only a limited set of 24 marker genes, which all overlapped with 10 dpi infected cell or wild type 5 dpi infected/primordia marker genes (Fig. 4d, **Source Data "5_WT_RH6"**).

***cyclops* mutants show very limited cortical response at 5 dpi**
In the *cyclops* samples, we identified only a few cells showing moderate expression of infection markers (Supplementary Fig. 8a), consistent with absence of cortical infection. To identify genes displaying *CYCLOPS*-dependent cortical expression patterns, we identified markers for infected wild-type cortical cells, comparing them against all *cyclops* cortical cells from rhizobia-inoculated plants within the same clusters (Fig. 4a, clusters 13, 16 and 26) and found 119 genes (**Source Data "5_C_NF-YA1"**), which, as expected, overlapped strongly with the 10 dpi infected and 5 dpi infected/primordia cell marker genes (Supplementary Fig. 8b).

The 5 dpi infected/primordia cells in the cortex were distributed across three different clusters (Fig. 4a, b, clusters 13, 16 and 26). We found no pronounced differences in nodulation gene expression between these clusters, which all comprised cells expressing *NF-YA1* and *NPL* (Supplementary Fig. 8a and c), suggesting that this pattern could be due to differences in cortical cell types rather than infection status. For instance, cluster 16 showed specific expression of a quiescence center marker (Supplementary Fig. 1c). Since the root tips, including the root apical meristem, were removed in the 5 dpi samples, these meristem-like cortical cell populations are likely associated with newly initiated nodule and/or lateral root primordia. To understand how closely related these nodule primordia cells were to the more mature nodule cells from the 10 dpi samples at the transcriptional level, we compared the 5 dpi cortical primordia marker genes to the 10 dpi infected, nodule and bacteroid markers. The uninfected nodule cell markers were largely unique, showing a relatively small overlap of 15 out of 260 genes with the 5 dpi markers (Fig. 4e). The 5 dpi overlap with the bacteroid cell markers of 42 out of 183 was larger, which is

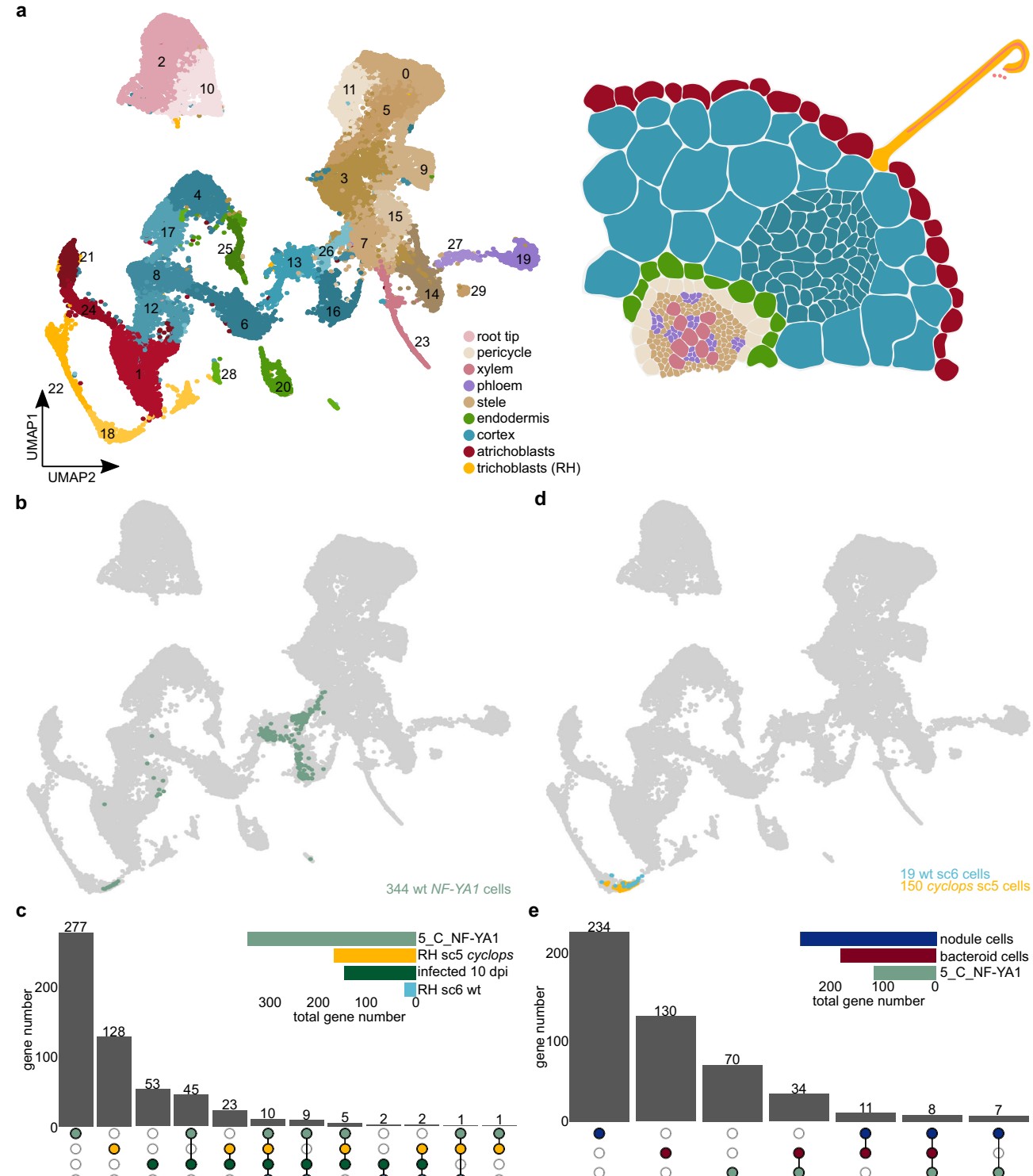

**Fig. 4 | *Cyclops* RH and cortex cells display unique features preventing IT formation. a** UMAP of wild-type (WT) and *cyclops* control and *M. loti* infected susceptible zone root cells 5 dpi. **b** UMAP of wt RH and cortical cells expressing *NF-YA1* (green) 5 dpi. **c** Upset plot comparing gene lists of RH and cortical cells expressing *NF-YA1* (5_NF-YA1) with RH subclusters (sc) 5 and 6 and the infected cells ten dpi (infected 10 dpi). **d** UMAP of wt sc6 (turquoise) and *cyclops* sc5 (yellow) RH cells expressing five dpi. **e** Upset plot comparing gene lists of RH and cortical cells expressing *NF-YA1* (5_C_NF-YA1) with nodule, bacteroid and infected cells 10 dpi (infected 10 dpi). Gene lists for all cell populations can be found in **Source Data**.

consistent with both marker lists being generated based on populations including infected cells. This again emphasizes the stability of the infection transcriptional module across time and cellular differentiation, and adds confidence to candidate gene identification.

**SYMRK-LIKE1 is required for normal infection thread formation**
Scrutinizing the infection-related candidate genes, we noticed that an apparently single-copy gene, *LotjaGi2g1v0191100*, annotated as a leucine-rich repeat receptor-like protein kinase, was among the top markers for 10 dpi infected cells. It showed higher specificity than NF-

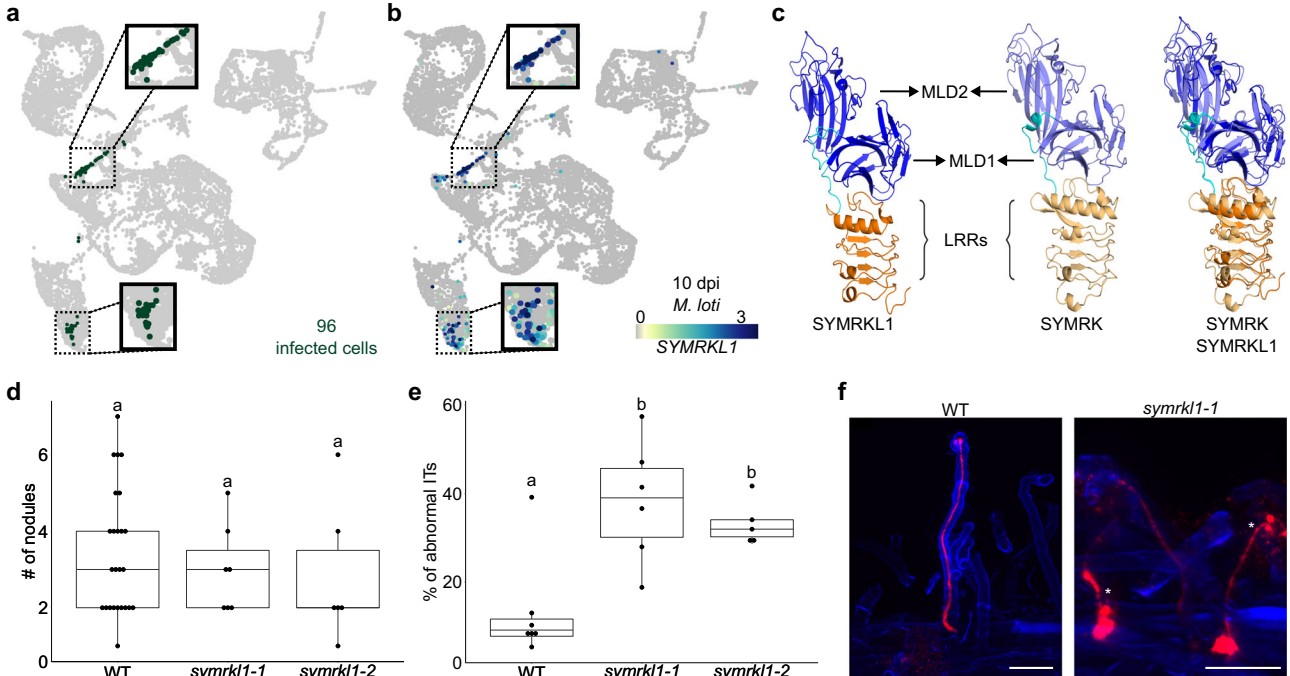

**Fig. 5 | SYMRKL1 is a l regulator of rhizobial infection. a** Illustration of infected cells in the 10 dpi UMAP. **b** Normalized expression of *SYMRKL1* in *M. loti* treated root hair and cortical cells 10 dpi. **c** Comparison of SYMRKL1 with SYMRK predicted ectodomains containing malectin-like domains (MLD) and leucine rich repeats (LRRs). **d** Nodule number and **e** percentage of abnormal infection threads (ITs) of wild-type and *symrkl1* plants 14 and 11 dpi. Letters indicate statistical groups (one-way ANOVA; $p \leq 0.05$; **d**: n(WT) = 25, n(symrkl1-1) = 7, n(symrkl1-2) = 6; e: n(WT) = 7, n(symrkl1-1) = 6, n(symrkl1-2) = 5). **f** Normal and abnormal wild type and *symrkl1-1* ITs. See also Supplementary Fig. 13. Abnormal ITs are marked by asterisks. Scale bar: 50 μm. Box plots in d and e show min, q1, median, q3 and maximum with outliers greater than 1.5x interquartile range shown individually. Exact *p* values can be found in **Source Data** "Fig. 5d raw and statistics", "Fig. 5e raw and statistics". Experiments in d and e have been successfully replicated twice.

YA1 (Fig. 5a, b, Supplementary Fig. 8a, b and 10, **Source Data "10_INF"**), but was not detected as a marker for the 10 dpi nodule or bacteroid cells (**Source Data "10_NOD" and "10_BAC"**). In addition, *LotjaGi2g1v0191100* was detected as a marker gene for 5 dpi infected/ primordia cells (Supplementary Fig. 10c, **Source Data "5_NF-YA1" and "5_C_NF-YA1"**) and was among the top markers for the 5 dpi wild type and *cyclops* responding root hairs (Supplementary Fig. 10c, **Source Data "5_Cyclops_RH5" and "5_WT_RH6"**). We confirmed the expression pattern of *SYMRKL1* in our dataset by transforming a triple YFP reporter construct into hairy roots (Supplementary Fig. 11). This highly consistent and infection-specific expression pattern suggests that *LotjaGi2g1v0191100* is involved in the infection process.

We then examined the *LotjaGi2g1v0191100* encoded protein sequence more closely, identifying a predicted tandem malectin-like motif (malectin-like domain, MLD) and three leucine-rich repeats (LRRs) sharing 136/545 (25%) amino acid identity with the ectodomain of Lotus SYMRK[19–21]. To determine the degree of structural similarity, we used AlphaFold[50,51] to model the two proteins. Despite the relatively low amino acid identity, the two models were nearly identical, with a superposition root mean square deviation value of 1.8 Å (Fig. 5c). We, therefore, named the protein SYMRK-LIKE1 (SYMRKL1). In contrast to the apparent similarity of their ectodomains, SYMRK contains a transmembrane- and intracellular kinase domain, whereas SYMRKL1 is attached to the membrane through a predicted GPI-anchor and lacks an intracellular kinase domain.

To functionally test the hypothesis that SYMRKL1 plays a role in rhizobium infection, we isolated two *LORE1 symrkl1* mutant alleles[45,52] and examined their infection phenotypes. The *symrkl1* mutants were indistinguishable from wild-type plants in terms of nodule number and nodule development (Fig. 5d, Supplementary Fig. 12). In contrast, the mutants showed large numbers of clearly aberrant infection threads

displaying various defects. These included enlarged bulbs, side branches and premature release of bacteria (Fig. 5e, f).

## Discussion

Forward genetic screens have been instrumental in the identification of major regulators with specific roles in nodulation. These approaches, however, require dramatic phenotypes that are easily identified in a background of many thousands of individuals and do not allow the identification of functionally redundant genes. Expression-based identification of nodulation genes also has challenges because of the very large set of genes affected directly or indirectly by rhizobium infection and nodule organogenesis. This effect was also evident in the single-cell transcriptomic study carried out in Medicago-infected roots, where more than 8000 differentially expressed genes were identified[41], and in the current study, making it challenging to prioritize candidates for experimental follow-up. Single-cell data further offers the opportunity to map developmental trajectories using pseudotime analysis, as was recently applied for indeterminate Medicago nodules[40]. Since Lotus forms determinate nodules while Medicago forms indeterminate nodules[7,8], future studies combining nodule single-cell RNA-seq data of both legume species could lead to a broader understanding of the mechanisms underlying the development of both nodule types. Pseudotime analysis is more challenging with respect to understanding the progression of rhizobial infection since each time point only captures a subset of the stages and because the infection process is superimposed on epidermal and cortical cells of different ages.

To leverage the single-cell resolution for high-confidence nodulation candidate gene identification, we focused on defining subsets of cells clearly linked to the nodulation process. Specifically, we targeted genes that showed expression patterns greatly enriched for the

subsets of nodulation-associated cells. The rationale is that such genes are unlikely to be required for general cellular functions and would instead be specifically linked to the genetic machinery required for successful nodulation. Indeed, this complement of specialized genes would be the focal point for potentially transferring nodulation and nitrogen fixation capacity to other plant species. Our gene lists, comprising more than 500 such genes, now provide a rich resource for further analysis, and their detailed expression patterns can easily be explored online through our shiny-app (https://lotussinglecell. shinyapps.io/lotus_japonicus_single-cell/).

Our approach does not capture all genes with nodulation-specific functions. Notably, the Nod factor receptors Nfr1 and Nfr5 did not pass our filtering criteria, likely because they are distributed across a wider set of cells in order to be available for the perception of rhizobial Nod factors prior to initiation of the infection process[53]. However, even the Nod factor receptors did show up as very high confidence markers for 5 dpi infected/primordia cells and *cyclops* responsive root hairs with *Nfr5* narrowly missing the specificity cutoff of 2% by being expressed in 2.5% of the remaining cells (**Source Data "5_NF-YA1" and "5_Cyclops_RH5"**). Since we have employed stringent thresholds for generating the gene lists presented in this study, we provide the unfiltered lists to allow more freedom in exploring the data (**Source Data**). The most striking transcriptional signature of the Nod factor receptors was their pronounced accumulation in responsive *cyclops* root hairs (Supplementary fig. 7c), indicating *Nfr* misregulation in the *cyclops* mutant.

In contrast to the *Nfrs, SYMRKL1* showed a very specific expression pattern, easily passing our filtering thresholds (Fig. 5a and Supplementary fig. 8). SYMRKL1 has an ectodomain very similar to that of SYMRK (Fig. 5c), and we have demonstrated the requirement of SYMRKL1 for normal infection thread progression (Fig. 5e, f). SYMRK is known to act upstream of $Ca^{2+}$ spiking upon Nod factor perception. Introducing the CCaMK gain-of-function mutation *snf1* into the nodulation- and infection-impaired *symrk* mutant[17,18] partially rescued the infection and nodulation phenotype, resulting in an increased number of misguided and malformed ITs[54]. Based on their structural similarity, it is tempting to speculate that the SYMRKL1 and SYMRK ectodomains interact with the same proteins and/or bind the same ligand and fine-tune infection thread formation, while the SYMRK kinase domain is required for nodule organogenesis. Indeed, cleavage of the MLD reduces SYMRK stability but enhances interaction with NFR5 and a mutation in the GDPC motif located in between SYMRK MLD and its LRRs impacts epidermal responses towards rhizobial infection in *symrk-14*[20,21]. A loss of the whole ectodomain, on the other hand, increases SYMRK stability[21]. As SYMRKL1 lacks a kinase domain, which is a crucial component of SYMRK function, the question remains how SYMRKL1 is involved in symbiotic signaling. One possibility is that SYMRKL1 acts as a decoy receptor as has been recently described in plants for the first time[55] and is common in mammals[56]. Another possibility is that SYMRK and SYMRKL1 form hetero-oligomers. One example of such hetero-oligomerization is the interaction of the GPI-anchored plasma membrane glycoprotein CHITIN ELICITOR BINDING PROTEIN (CEBiP) and CHITIN ELICITOR RECEPTOR KINASE (CERK1) during chitin perception in rice[57]. Given that *symrkl1* plants display no significant difference in nodule number or maturation, our data provide a valuable resource for identifying nodulation genes, especially those whose loss causes mild phenotypes or are subject to functional redundancy.

## Methods
### Plant material
*Lotus japonicus* seeds from the Gifu accession (both WT and mutants) were scarified with sandpaper, sterilized with 1% (v/v) sodium hypochlorite for 10 min and then washed 5 times with sterile water under sterile conditions. The seeds were incubated overnight at 4 °C and then transferred to square Petri dishes for germination under a 16 h day (at 21 °C) and 8 h (at 19 °C) night cycle. After three days seeds with emerging radicles were transferred to square plates with 1.4% Agar Noble slopes containing 0.25x B&D medium and covered with filter paper. A metal bar with 3-mm holes for roots was inserted at the top of the agar slope. Plant growth plates, each containing 10 seedlings, were inoculated with 500 μL of OD600 = 0.02–0.05 bacterial suspensions along the length of the root. For genetic studies, *LORE1* lines *symrkl1-1* (30085537) and *symrkl1-2* (30090169)[52] as well as *cyclops-2* were used[27].

### Rhizobia Strain
The *M. loti* R7A rhizobia strain was used for *L. japonicus* nodulation. Rhizobia was cultured for 2 days at 28 °C in yeast mannitol broth (YMB).

### Protoplast Isolation and scRNA-seq
For protoplast isolation, whole roots or susceptible zones were protoplasted under slight shaking for 3 h at room temperature in 5 mL digestion solution (10 mM MES (pH 5.7), 1.5 % (w/v) cellulase R-10, 2 % (w/v) macerozyme R-10, 0.4 M D-sorbitol, 10 mM $CaCl_2$, 5 % (v/v) viscozyme, 1 % (w/v) BSA. Intact protoplasts were isolated by filtering the protoplast-containing digestion solution with a 40 μM strainer into 15 mL falcon tubes and mixing it with 5 mL of 50 % Optiprep solution (50 % (v/v) Optiprep,10 mM MES (pH 5.7), 0.4 M D-sorbitol, 5 mM KCl, 10 mM $CaCl_2$) and topping the mixed solution carefully first with 2 mL of 12.5 % Optiprep (12.5 % (v/v) Optiprep,10 mM MES (pH 5.7), 0.4 M D-sorbitol, 5 mM KCl, 10 mM $CaCl_2$) and 250 μL of 0 % Optiprep (10 mM MES (pH 5.7), 0.4 M D-sorbitol, 5 mM KCl, 10 mM $CaCl_2$). Falcon tubes were centrifuged for 10 min at 250 g at 4 °C. Living protoplasts were collected at the interphase of 12.5 and 0 % Optiprep solution and counted with a Neubauer counting chamber. For scRNA-seq library preparation, the Chromium Next GEM Single Cell 3' Kit v3.1 was used following the manufacturer's protocol aiming to recover 5000 cells per biological replicate.

### Bulk RNA-seq
To identify protoplast-induced genes, we protoplasted whole roots grown as for the 10 dpi dataset (see "Protoplast Isolation and scRNA-seq"). At the time of protoplast harvest, we flash-froze intact whole roots in liquid nitrogen. We isolated RNA using the NucleoSpin RNA Plant Kit (Macherey-Nagel). Libraries were constructed[58] and bulk RNA-seq were performed by Novogene (UK).

### Computational analysis of single-cell transcriptomes
**Raw data pre-processing, integration and clustering.** Raw sequencing data were processed using Cell Ranger v6.1.2 (10X Genomics). As reference for "cellranger mkref" the *Lotus japonicus* Gifu v1.2 and Gifu v1.3 were used for genome assembly and gene annotations, respectively (available in Lotus Base, lotus.au.dk[45]. "cellranger count" was run with the default parameters using STAR v2.7.2a[59] as the aligner. The "filtered_feature_bc_matrix" was used as input for the next steps.

The downstream analyses were carried out using Seurat 4.0.5[44]. The Cell Ranger matrices were further filtered to eliminate low-quality cells and genes. Specifically, any cells that had less than 200 or more than 7500 expressed genes and less than 500 UMIs were eliminated from the analysis. The cells were next filtered based on the mitochondrial and chloroplast encoded gene expression, retaining only the cells expressing under 5% read counts from these features in the 10 dpi. For the 5 dpi dataset, the cells expressing under 10% mitochondrial genes and 5% chloroplast genes were retained. Additionally, only genes that were expressed in at least three cells were included in the analysis.

All samples were normalized using the "sctransform" function implemented in Seurat, with "vars.to.regress" set to mitochondrial and chloroplast genes[60]. For the 5 dpi dataset, the method = "glmGamPoi"[61]

was used. The samples were integrated using the canonical correlation analysis integration pipeline from Seurat, with the Control datasets used as reference for the 10 dpi samples and without a reference dataset for the 5 dpi samples. The number of integration anchors was set to 3000.

Using the integrated data assay, PCA dimensionality reduction was run using the default function in Seurat. Then, the functions "FindNeighbors" and "RunUMAP" were run using 50 principal components for the full datasets and 30 for the root hair subsets. The cells were clustered using the unsupervised Lovain clustering algorithm with the default resolution of 0.8.

ShinyCell[62] was used to create a web interface for visualizing the single-cell datasets (https://lotussinglecell.shinyapps.io/lotus_japonicus_single-cell/).

**Differential gene expression and marker identification.** To detect genes induced by protoplasting, fastp[63] (v 0.23.4) with the flags -g --low_complexity_filter -q 30 was used to trim the forward reads of all the samples. Next, the forward reads were mapped to the *L. japonicus* genome using STAR (v2.7.2a) with the parameters --alignIntronMax 10,000 and --quantMode GeneCounts. The gene counts generated by STAR were used for the differential gene expression analysis using the edgeR[64] (v3.34.1) available in R. After filtering out the lowly expressed genes, the counts were normalized using the Trimmed Mean of M-values method. We tested for DE genes between the protoplasts and whole root samples in the uninoculated condition using the quasi-likelihood F-test method. The genes were selected as differentially expressed if they had a $\log FC \geq |2|$ and an $FDR \leq 0.001$.

Gene expression was normalized using the NormalizeData function on the Seurat RNA assay. This assay was then used for the differential gene expression analyses and plotting. Gene markers specific for each cell cluster were identified using the "FindConservedMarkers" function with the Wilcoxon rank-sum test method, with the "grouping.var" = "Treatment" for the 10 dpi dataset and "Sample" for the 5 dpi dataset. In the ten dpi dataset, after filtering for a "pct.2" <0.1 in both conditions, we selected the three markers with the highest logFC for each cluster (Fig. 1b).

Differentially expressed genes between treatments for each cluster and the markers for specific groups of cells were identified using the "MAST" algorithm v 1.16[65] implemented in Seurat "FindMarkers", with the "min.pct" = 0.01 parameter. The genes with an adjusted $p$ value $\leq 0.05$ and a log fold change $\geq |0.25|$ were considered differentially expressed. Additional filtering settings for each of the gene lists are detailed in **Source Data**.

**Scissor algorithm.** The Scissor software v 2.0.0 (Sun at al.[49]) was used on the subset of root hairs clusters in conjunction with the raw reads from a published set of root hair bulk RNA-seq data (Kelly et al.[38]) using the family = "binomial" and alpha=NULL parameters.

**Promoter reporter constructs**
Promoter fragments (~2 kb) of selected marker genes identified by single-cell RNA Sequencing were either amplified using specific primers carrying *BsaI* sites overhangs and cloned in the Golden Gate compatible vectors to generate either promoter:GUS (β-glucuronidase) or entry vectors containing promoter fragments were synthesized by Thermo Fisher and subsequently used for cloning promoter:tYFP constructs.

**Hairy root transformation**
pIv10 vectors harboring promoter:GUS constructs were integrated into the pRI (root-inducing plasmid) of *Agrobacterium rhizogenes* AR1193 by homologous recombination using the helper *E. coli* strain GJ23. Hairy root formation was induced by piercing Lotus seedlings at the hypocotyl site by using a narrow needle with a drop of

agrobacterium in YMB medium. Three weeks after inoculation with *Agrobacterium* the primary root of infected seedlings was cut, and plants exhibiting hairy roots were placed into lightweight expanded clay aggregate (LECA) and inoculated with *M. loti* to promote nodulation.

**Histochemical analysis of GUS activity**
Transformed hairy roots carrying promoter:GUS vectors were immersed in 5-bromo-4-chloro-3-indolyl-β-D-glucuronic acid (X-Gluc 0.5 mg/ml) containing solution (100 mM $NaPO_4$ pH 7.0, 10 mM EDTA pH 8.0, 1 mM K Ferricyanide, 1 mM K Ferrocyanide), and vacuum-infiltrated for 10 min. Histochemical staining for GUS activity was performed at 37 °C for 12 h. After staining, the roots were fixed in 70% ethanol and embedded in 2,5% agar. Then, roots were transversely and/or longitudinally cut in 80 μm sections with a vibratome (Leica VT1000 S). GUS activity was observed with a light microscope (Zeiss AxioPlan2) equipped with a camera.

**Confocal microscopy**
Confocal microscopy was performed with Zeiss LSM780 microscope. Following excitation/emission [nm] settings were used: (i) auto-fluorescence of cell components 405/420–505, (ii) DsRed 561/580–660.

**Structural analysis of SYMRK proteins**
Predicted models of Lotus SYMRK ectodomain (aa 30-509) and Lotus SYMRKL1 (aa 25-480) were generated through the Alpha-Fold2_MMseqs2 implementation of AlphaFold2 at ColabFold[66]. Default settings were utilized (use_amber: no, template_mode: none, msa_mode: MMSeqs2, pair_mode: unpaired+paired, model_type: auto, num_recycles: 3). 5 nearly identical models were created for each protein, with the top pLDDT ranked models being chosen for further structural analysis in PyMOL (The PyMOL molecular graphics system, version 2.5.2, Schrödinger, LLC).

**Statistical analysis for *symrkl1* mutants**
Statistical analysis was performed with SAS®Studio (SAS Compliance Department, NC, USA; https://odamid.oda.sas.com/SASStudio, last accessed on 1 November 2022). Homogeneity and homoscedasticity were tested by Shapiro–Wilk ($p \geq 0.95$, $p \leq 0.95$ but $Pr <W \geq 0.05$) and Levene tests ($p \geq 0.01$) before ANOVA testing was performed followed by Tukey post-hoc test. For analysis of nodule numbers, which did not meet the assumptions initially, log10 transformation was performed.

**Reporting summary**
Further information on research design is available in the Nature Portfolio Reporting Summary linked to this article.

## Data availability
The sequencing data generated in this study have been deposited as ENA project accession PRJEB57790. UMAPs and gene expression data can be browsed at [https://lotussinglecell.shinyapps.io/lotus_japonicus_single-cell/]. The Seurat objects and metadata tables generated in this study have been deposited as Supplementary Data at Figshare [https://doi.org/10.6084/m9.figshare.23986200.v2]. All generated gene lists, nodulation and infection thread data, and promoter sequences are provided as **Source Data**. Source data are provided with this paper.

## Code availability
The codes used for data analyses in this study have been deposited on GitHub under the following link: [https://github.com/LaviFechete/Lotus_Single_Cell] and has been assigned a doi [https://doi.org/10.5281/zenodo.8435525].

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

## Acknowledgements

This work was supported by the European Research Council (ERC) under the European Union's Horizon 2020 research and innovation programme (grant agreement no. 834221) and by Independent Research Fund Denmark (grant agreement no. 1026-00032B).

## Author contributions

M.F., F.T., D.R. and S.U.A. conceived and designed experiments; M.F., F.T. and D.R. performed experiments; L.I.F. conducted bioinformatic scRNAseq analysis and set up the shiny app, supervised by S.U.A. and M.H.S.; M.F., L.I.F., F.T., D.R. and S.U.A. analyzed the data; J.M. provided plant materials; M.N. performed microscopy; M.M.M.N. and K.R.A. performed structural analysis; M.F. drafted the first version of the manuscript; S.U.A., M.F. and D.R. edited the manuscript with input from all authors.

## Competing interests

The authors declare no competing interests.
