## [Peer Review File · Nature Communications]

Single-cell analysis identifies genes facilitating rhizobium infection in *Lotus japonicus*REVIEWER COMMENTS

Reviewer #1 (Remarks to the Author):

The manuscript entitled, “Single-cell analysis maps distinct cellular responses to rhizobia and identifies the novel infection regulator SYMRKL1 in *Lotus japonicus*” by Frank, et al. describes a set of single-cell experiments on *Lotus* plants infected (or uninfected) with rhizobia, including a mutant, cyclops. This is a really exciting work that applies a powerful technology to a very interesting biological question: what are the various mechanisms within cells that mediate a complex symbiosis. While (understandably) the authors focus on hair and cortex cells for their study, the datasets themselves should prove of great use to studying other plant responses to rhizobia in a cell type-specific way. I do have a few, mostly minor comments that I hope could improve the manuscript.

I did not find a cell-level metadata table containing cell identifiers with the imputed information (cell type, sample of origin, infection status, cell classifications, etc.). This is critically important for the reuse of your analyses and should be included somewhere (not sure if this is possible at the ENA, but at the very least could add another supplemental table).

While replication was performed for the 10 dpi experiment, the authors do not describe roughly how similar they are (though I expect there will not be any surprises).

I was wondering why the authors selected 10 dpi as a time point for the WT study, but 5 dpi for the cyclops study. While I understand that cyclops mutants arrest nodule development early, does this have other detrimental impacts if kept for longer? Having the time point be the same could potentially enable all the datasets to be more evenly compared. Please provide a rationale in the text to justify this choice (unless I missed it).

Did the authors attempt to integrate all the single-cell data in their study together into a combined dataset? This might make some of the comparisons a little more straightforward.

How evenly are cells from different treatment groups distributed within clusters? I noticed from the single-cell visualization tool (shiny app) that there are a couple of clusters that are dominated by one sample versus another. Any thoughts as to what these represent?

L127-128 – “we identified 1 cells from cluster 8 as bacteroid cells based on their expression of leghemoglobin genes LB1, LB2, or LB3” – only expression of LB1 is shown.

In figure 2g, I would consider directly annotating the cluster of cells with high NPL expression (i.e. call out with a box), as it's easy to miss them as the other cells take up so much space.

L154 – “we split the 96 infected cells by tissue, identifying 27 infected root hair .. cells” – does this correspond well with the 36 uninfected RH cells identified in the previous section? Or is there a distinction between rhizobium response and direct infection being drawn here? Some clarity could be provided.

In calling out subclusters 5 and 6 in the 5dpi dataset, the authors point to a differential enrichment between wild-type and cyclops infected vs. non-infected, but I did not see this quantified anywhere. From looking at the UMAP, it seems as though cluster 6 is also enriched in infected cyclops cells. Please clarify.

Last, in the methods, the authors state that different thresholds were employed for mitochondrial/chloroplast gene percentages as a quality filter. Was there a rationale behind this (i.e. did one dataset have a much higher fraction of organelle reads than the other) or is it some artifact of an early analysis discrepancy? Either way I think it's ok (I doubt this would change the analysis by much), but if there is some quality issue with either dataset warranting a more stringent filter, this would be good to know.

Reviewer #2 (Remarks to the Author):

Frank, Fehete, and co-authors use droplet-based scRNA-seq to profile protoplasts from rhizobia-inoculated and mock-treated WT and mutant Lotus roots. From the integrated 10 days post-inoculation (dpi) dataset, major root cell types were annotated based on the expression of known markers. The primary cell populations of interest, including bacteroid-containing, nodule, and infected cells, were then identified from the inoculated data. To investigate differences in gene expression programs activated by infected cortical versus root hair cells, the authors identified modules of genes expressed in each infected cell type and also in both cell types. To further differentiate between infected root hair and cortical programs, the authors generated scRNA-seq data from 5dpi WT and cyclops mutant roots. Based on comparisons between the WT 5dpi and 10dpi datasets, a main conclusion is that similarities in gene expression indicate stability across time for the infection transcriptional module. Finally, phenotyping infected *symrkl1* mutants led to the conclusion that SYMRKL1 is required for normal infection thread formation.

I am an experienced scRNA-seq user but not a nodulation expert. Overall, the scRNA-seq analysis follows the standard Seurat pipeline that has been used for previous plant datasets. However, some details, useful data visualizations, and analyses are missing and should be addressed to bring this study on par with the existing literature. A specific concern is the lack of *in vivo* validation of the three cell populations of interest in the inoculated samples, particularly given that the numbers of cells are very small. 21 bacteroid cells, 98 nodule cells, and 96 infected cells were identified out of ~12,000 cells from the inoculated dataset (10dpi). Identification of these populations in this study is based only on *in silico* analyses. Previously published images of only 1 promoter:GUS/GFP reporter line for each population are shown in Supplementary Figure 2. Further, the identification of the same cell populations in the 5dpi WT and cyclops datasets is validated in several instances based on comparisons with the 10dpi data. Given that this work seems to represent the first droplet-based scRNA-seq data from Lotus (lines 82-91), I think it is important to further validate the annotation by adding *in vivo* expression data for some of the newly identified *in silico* markers, at a minimum for 10dpi WT roots.

Line 100: Protoplasting induces stress and thus should be taken into account in scRNA-seq data analysis. This seems especially important given the relatively long 3-hour digestion time used here for Lotus roots (Methods). Were protoplasting-induced genes removed from the data? This is a common procedure in analyses of scRNA-seq data from Arabidopsis roots.

Line 101: Can the authors explain the rationale behind deciding the number of cells to profile with scRNA-seq, especially given the low frequency of detecting bacteroid, nodule, and infected cells?

Line 103: Please indicate in the main text how many cells of the 25,024 are from the 10dpi inoculated samples and how many are from the mock-treated samples (this information is only found in the last two tabs of the supplemental file). In the supplemental file, it would be also be useful include the number of cells annotated for each root cell type from each of the treatment groups.

In addition to the integrated UMAP in Fig 1a that is colored by cell type annotation, the authors should show UMAP visualizations to facilitate comparison of the mock-treated versus inoculated samples. For example, the UMAP plot of the integrated dataset can be colored according to treatment type (I see that this plot is available on the Shiny app but it should be included in the manuscript also). The two treatment groups can also be colored on two separate UMAP plots while preserving the coordinates of the integrated dataset. To plot cell type annotations from each treatment group separately with Seurat, use the 'split.by' parameter and set the 'group.by' parameter to the cell type annotation.

Lines 103-104: Clustering requires user-selected parameters and the number of clusters changes based on the parameters. How does the recovery of 32 clusters relate to the expected number of cell types? For example, in Figure 1a, trichoblast cells are split into two clusters. Why? Is one cluster from the inoculated sample and one from the mock-treated sample? Similarly, for the cortex clusters, do the different layers of cortex have different transcriptional profiles? This also relates to my comment above about the utility of additional UMAP visualizations.

An alternative cell type annotation approach is to first use cell type specific markers to annotate the mock-treated control sample. Then, the label transfer function in Seurat can be used to annotate the inoculated dataset. With this approach, it should be possible to identify cell types in the inoculated dataset that are absent in the mock-treated sample. Can the authors clarify why they chose to assign cell types based on the integrated inoculated and mock-treated data? This approach requires the assumption that inoculation does not substantially alter root cell identities. Do the nodule and bacteroid cells not constitute unique cell types?

Line 107: Do the proportions of the recovered cell types in the scRNA-seq data match the proportions of Lotus root cell types in vivo?

Line 109: Can the authors use a more quantitative metric to explain what is a 'substantial' transcriptional response to infection? In Figure 1C, neither the plot itself nor the legend indicate the cutoff for a gene to be considered differentially expressed (e.g., log fold change of 2?).

Line 152, section entitled 'Root hair and cortical infection transcriptional programs differ.' The title of this section in the main text suggests that it is the unique programs that are noteworthy. However, the title of the Figure 3 legend (line 473; 'Root hair- and cortex-IT cells share common gene sets') highlights the opposite finding. Can the authors clarify if one of these results is of particular interest and why?

Line 171: Of the 32,180 high quality cells, please indicate in the main text how many are from the cyclops mutant and WT genotypes with mock treatment or inoculation, respectively.

Lines 242-247: I don't quite understand the big-picture significance of the symrkl1 phenotype. Is the mutant unable to fix nitrogen?

Line 435, Data availability: I encourage the authors to make the final Seurat objects available for download in addition to the sequencing data. This prevents the need for community members to re-analyze the raw data in order to explore it further. Although the scRNA-seq analysis-related methods are reasonably thorough, it is helpful to publish all code (e.g., via GitHub) and I encourage the authors to do this for reproducibility. I like the Shiny app as a resource for researchers who are interested in visualizing the final datasets. It is user-friendly and has a large amount of available information.

Minor comments

Line 101: The authors indicate that they carried out scRNA-seq of protoplasts from mock-treated and rhizobium-inoculated Lotus roots 'in duplicates.' Based on the methods (line 341), the duplicates are specifically biological replicates. Please clarify this detail in the main text. Further, can the authors clarify in the main text that 25,024 is the final number of cells after filtering out low quality cells.

Figure 1 b and c: The cluster numbers are very small and hard to see, especially when on top of the very small colored boxes. The dots in the dot plot are also very small and difficult to see.

Figure 1c: In the legend, please indicate if the genes are up-regulated in the infected sample relative to the control or vice versa.

Line 106 and Supplemental Figures 1 and 2: The figure legends for Supp Figures 1 and 2 indicate that scRNA-seq data are shown for both 10dpi and 5dpi samples. However, at the first mention of these supplemental datasets in the main text (line 101), only 10dpi samples have been described in detail. I assume that the 5dpi samples in these figures are from the WT controls run alongside the cyclops mutant. Please clarify this in the main text and figure legends.

Line 110: The least responsive tissues were phloem, QC, and xylem. Can the authors clarify if this finding was expected for xylem and phloem given their location deep in the center of the root?

Line 519, Supp Fig 3: The cell type annotation legend for the cluster numbers and colors should be included as in Fig 1a.

Reviewer #3 (Remarks to the Author):

This is a very impressive single cell analysis of the transcriptome on the various cell types making up a *L. japonicus* (determinate) nodule. I found the work to be highly novel and a major piece of work for the field of nodulation. The identification of *symrk11* from its expression in a cell specific manner and the absence of its expression in a *Cyclops* mutant is extremely impressive and novel. That *Symrk11* is an important protein is clear from the aberrant infection threads formed by two independent *Lore1* mutants of *symrk11*. It is clear that traditional forward screens would not have identified the role of *SYMRK11* as the mutation does not alter an easily screened phenotype such as reduced nodule number or nitrogen starvation resulting from impaired N₂ fixation. My one suggestion to the investigators relates to the absence of a nodulation phenotype in the *Lore1* mutants. As far

as I can tell 0.5 ml of a 0.02-0.05 OD600 culture of Mesorhizobium lotus R7A were used as an inoculum. As a rule of thumb most rhizobia have approx 10^9 cells per ml at OD600 1, so 0.5 ml of a 0.02 OD600 culture has approx 2×10^7 cells and OD0.05 has approx 2.5×10^7 bacterial cells. These are staggering high inocula and microbiologists working in competition assays would inoculate 10^3 - 10^4 rhizobia on each plant. My point is if the nodulation experiments were done with 10^3 - 10^4 R7A per plant there may well be a difference in final nodule number or in the timing of nodule formation (i.e. nodule appearance over time) between wildtype and symrkl1 mutants. This would show why the role of SYMRKL1 is critical in the real work of soil in the field with natural numbers of infecting rhizobia. Of course there may be no difference but it would be the cherry on top of what is already a great piece of work.

We wish to thank the reviewers for their careful evaluation of our work and for their constructive comments, which were very helpful in our efforts to improve the manuscript. Please find our responses to all comments in blue below.

REVIEWER COMMENTS

Reviewer #1 (Remarks to the Author):

The manuscript entitled, “Single-cell analysis maps distinct cellular responses to rhizobia and identifies the novel infection regulator SYMRKL1 in *Lotus japonicus*” by Frank, et al. describes a set of single-cell experiments on *Lotus* plants infected (or uninfected) with rhizobia, including a mutant, cyclops. This is a really exciting work that applies a powerful technology to a very interesting biological question: what are the various mechanisms within cells that mediate a complex symbiosis. While (understandably) the authors focus on hair and cortex cells for their study, the datasets themselves should prove of great use to studying other plant responses to rhizobia in a cell type-specific way. I do have a few, mostly minor comments that I hope could improve the manuscript.

I did not find a cell-level metadata table containing cell identifiers with the imputed information (cell type, sample of origin, infection status, cell classifications, etc.). This is critically important for the reuse of your analyses and should be included somewhere (not sure if this is possible at the ENA, but at the very least could add another supplemental table).

That is a good point and we are happy to supply the requested data. We have made the tables available on figshare together with the Seurat objects: <https://doi.org/10.6084/m9.figshare.23986200.v2>. This information is included in the manuscript under “Data availability”

While replication was performed for the 10 dpi experiment, the authors do not describe roughly how similar they are (though I expect there will not be any surprises).

We added the following description in lines 107 to 110: “We assessed the similarity of the samples by correlating the gene counts in the RNA assay after normalization. Control replicates had a Pearson correlation factor of 0.99 and inoculated replicates of 0.98 (**Supplemental file 1 "Correlation 10dpi"**). The Pearson correlation coefficient between the control and the inoculated samples was in the 0.88 to 0.90 range.” We have included an overview here for clarification.

I was wondering why the authors selected 10 dpi as a time point for the WT study, but 5 dpi for the cyclops study. While I understand that cyclops mutants arrest nodule development early, does this have other detrimental impacts if kept for longer? Having the time point be the same could potentially enable all the datasets to be more evenly compared. Please provide a rationale in the text to justify this choice (unless I missed it).

Yes, we now realize that this was not clearly stated. The rationale is that cyclops mutants display early arrest of infection, already before infection threads elongate. To compare the aborted infection events in cyclops to wild type infection threads that have not yet fully traversed the root hair cells, we chose the 5 dpi timepoint. At 10 dpi, most infection threads would already be fully elongated within root hairs, either arresting at that point or continuing to proliferate in cortical cells. We have now elaborated on this in the text in lines 186-187.

Did the authors attempt to integrate all the single-cell data in their study together into a combined dataset? This might make some of the comparisons a little more straightforward.

Yes, initially we integrated all datasets into a single object. However due to differences in plant age and tissue developmental stages, the integrated data tended to cluster by age making it less amenable to capturing the infection-related signals we were interested in.

How evenly are cells from different treatment groups distributed within clusters? I noticed from the single-cell visualization tool (shiny app) that there are a couple of clusters that are dominated by one sample versus another. Any thoughts as to what these represent?

Yes, that is correct and it is related to the effects of rhizobium infection. Cell populations containing infected, nodule and bacteroid cells (clusters 14 and 22) and cells that are responsive to rhizobium infection, e.g. epidermis (cluster 4), have a higher proportion of cells in the treated samples. Cell types with no or little response to rhizobia are overrepresented in the control samples. To clarify these effects, we have added a new supplemental table summarizing the cell counts across samples, clusters and tissue types in Supplemental file 1 “10 dpi Cells_cluster”. We have clarified these effects in lines 122 to 125.

L127-128 – “we identified 1 cells from cluster 8 as bacteroid cells based on their expression of leghemoglobin genes LB1, LB2, or LB3” – only expression of LB1 is shown.

While we have exemplified the expression pattern of LB1 in the figure, all LB genes are expressed in bacteroid-containing cells (see Fig.1 of Wang et al., 2019). We have added references to the respective publications and the expression patterns to our validated gene list (Supplemental file 1 “VAL”).

In figure 2g, I would consider directly annotating the cluster of cells with high NPL expression (i.e. call out with a box), as it’s easy to miss them as the other cells take up so much space.

We have included a gray arrow pointing to the positively responding cell population in 2f and a blue arrow pointing to the NPL expressing population in 2g accordingly. We added the following sentences to the figure description: “The cell population with the highest proportion of positively responding cells is indicated by a gray arrow.” and “The cell population with the highest proportion of *NPL* expressing cells is indicated by a blue arrow.”

L154 – “we split the 96 infected cells by tissue, identifying 27 infected root hair .. cells” – does this correspond well with the 36 inefected RH cells identified in the previous section? Or is there a distinction between rhizobium response and direct infection being drawn here? Some clarity could be provided.

The 27 cells are a subset of the 36 cells identified in the previous section. The scissor algorithm marked the cluster of cells responding to inoculation as positive, as well as 9 cells in other root hair clusters. Since these 9 cells do not cluster with the other 27 and were not identified by the second approach, we did not include them in the downstream analysis. We have clarified this in lines 173-174.

In calling out subclusters 5 and 6 in the 5dpi dataset, the authors point to a differential enrichment between wild-type and cyclops infected vs. non-infected, but I did not see this quantified anywhere. From looking at the UMAP, it seems as though cluster 6 is also enriched in infected cyclops cells. Please clarify.

Thank you for the comment. It is correct that subcluster 6 also shows an enrichment of cyclops inoculated cells, although the tendency is not as strong as for subcluster five, and subcluster 6 contains much fewer cells. We have added a supplemental table quantifying the number of cells from each sample within the Root hair subclusters in Supplemental file 1 “5 dpi RH subclusters”.

Last, in the methods, the authors state that different thresholds were employed for mitochondrial/chloroplast gene percentages as a quality filter. Was there a rationale behind this (i.e. did one dataset have a much higher fraction of organelle reads than the other) or is it some artifact of an early analysis discrepancy? Either way I think it's ok (I doubt this would change the analysis by much), but if there is some quality issue with either dataset warranting a more stringent filter, this would be good to know.

Different thresholds were used for the organelle reads in the two datasets because the quality was different. The 5dpi data set had higher counts for the mitochondrial and chloroplast reads. This also contributed to our decision to analyze the 10 and 5 dpi datasets separately as described above.

Reviewer #2 (Remarks to the Author):

Frank, Fehete, and co-authors use droplet-based scRNA-seq to profile protoplasts from rhizobia-inoculated and mock-treated WT and mutant Lotus roots. From the integrated 10 days post-inoculation (dpi) dataset, major root cell types were annotated based on the expression of known markers. The primary cell populations of interest, including bacteroid-containing, nodule, and infected cells, were then identified from the inoculated data. To investigate differences in gene expression programs activated by infected cortical versus root hair cells, the authors identified modules of genes expressed in each infected cell type and also in both cell types. To further differentiate between infected root hair and cortical programs, the authors generated scRNA-seq data from 5dpi WT and cyclops mutant roots. Based on comparisons between the WT 5dpi and 10dpi datasets, a main conclusion is that similarities in gene expression indicate stability across time for the infection transcriptional module. Finally, phenotyping infected *symrkl1* mutants led to the conclusion that SYMRKL1 is required for normal infection thread formation.

I am an experienced scRNA-seq user but not a nodulation expert. Overall, the scRNA-seq analysis follows the standard Seurat pipeline that has been used for previous plant datasets. However, some details, useful data visualizations, and analyses are missing and should be addressed to bring this study on par with the existing literature. A specific concern is the lack of in vivo validation of the three cell populations of interest in the inoculated samples, particularly given that the numbers of cells are very small. 21 bacteroid cells, 98 nodule cells, and 96 infected cells were identified out of ~12,000 cells from the inoculated dataset (10dpi). Identification of these populations in this study is based only on in silico analyses. Previously published images of only 1 promoter:GUS/GFP reporter line for each population are shown in Supplementary Figure 2. Further, the identification of the same cell populations in the 5dpi WT and cyclops datasets is validated in several instances based on comparisons with the 10dpi data. Given that this work seems to represent the first droplet-based scRNA-seq data from Lotus (lines 82-91), I think it is important to further validate the annotation by adding in vivo expression data for some of the newly identified in silico markers, at a minimum for 10dpi WT roots.

As suggested, we have conducted additional cluster validation and added confocal images of hairy roots expressing *LotjaGi2g1v0052900:tYFP* (nodule cell marker, weak cortex expression) and *LotjaGi1g1v0258800:tYFPnls* (pericycle marker) to Supplemental Figure 1. The *LotjaGi1g1v0258800:tYFPnls* data led us to revise the tissue type annotation for cluster 6 in the 10 dpi dataset and cluster 11 in the 5 dpi dataset to pericycle. We thank the reviewer for this suggestion, which has helped improve the accuracy of our annotation.

We have also added confocal images showing *SYMRKL1:tYFPnls* expression in Supplemental Figure 10 (infected root hair and infected cortex expression).

In addition to the promoter-reporter data we provide here, our identification of infected, nodule and bacteroid-containing cell populations is based on several well-characterized genes. We realize now that this was not sufficiently clear and have therefore amended the “Experimentally validated nodulation genes” list, adding the columns “reference”, “in vivo expressed in”, “expression 10 dpi” and “expression 5 dpi”. We have also clarified this in the main manuscript text in lines 139-140.

Line 100: Protoplasting induces stress and thus should be taken into account in scRNA-seq data analysis. This seems especially important given the relatively long 3-hour digestion time used here for Lotus roots (Methods). Were protoplasting-induced genes removed from the data? This is a common procedure in analyses of scRNA-seq data from Arabidopsis roots.

To address this concern, we have performed a bulk RNA-seq experiment to identify protoplast-induced genes. We identified 655 genes that were induced by protoplasting in

Lotus japonicus, and we have marked these in the gene lists as “protoplast-induced”. We find that just one protoplast-induced gene was included in the gene lists associated with the infected, nodule or bacteroid-containing cells at 10 dpi, indicating that protoplasting has only minimal impacts on the symbiotic cell populations we have focused on. In a recent study in the legume *Medicago* (Ye et al 2022), protoplasting was performed for 2.5 h, which is comparable to protoplasting protocol used here. We have commented on this in lines 124-127. We have also added a description of sample collection for this experiment in the Methods section under “Bulk RNA-seq” (lines 377-382).

Line 101: Can the authors explain the rationale behind deciding the number of cells to profile with scRNA-seq, especially given the low frequency of detecting bacteroid, nodule, and infected cells?

The number of cells analyzed in each run was partly determined by the technical recommendations of the 10X platform (to avoid overloading and cell doublets) as well as the number of cells of interest detected in our initial experiments. The highly specific expression profile associated with infected cells allowed us to identify the associated genes with high confidence and provided a large number of new genes and biological processes of interest for further genetic analyses.

Line 103: Please indicate in the main text how many cells of the 25,024 are from the 10dpi inoculated samples and how many are from the mock-treated samples (this information is only found in the last two tabs of the supplemental file). In the supplemental file, it would be also be useful include the number of cells annotated for each root cell type from each of the treatment groups.

We have indicated the number of cells from the mock and treated samples for the 10 dpi dataset in the main text in lines 104-105. The table quantifying the number of cells from each cell type can be found in Supplemental file 1 “10 dpi Cells_Cell type”, which we refer to in the text in line 106.

In addition to the integrated UMAP in Fig 1a that is colored by cell type annotation, the authors should show UMAP visualizations to facilitate comparison of the mock-treated versus inoculated samples. For example, the UMAP plot of the integrated dataset can be colored according to treatment type (I see that this plot is available on the Shiny app but it should be included in the manuscript also). The two treatment groups can also be colored on two separate UMAP plots while preserving the coordinates of the integrated dataset. To plot cell type annotations from each treatment group separately with Seurat, use the ‘split.by’ parameter and set the ‘group.by’ parameter to the cell type annotation.

We have added the requested plots to Supplemental Figure 4.

Lines 103-104: Clustering requires user-selected parameters and the number of clusters changes based on the parameters. How does the recovery of 32 clusters relate to the expected number of cell types? For example, in Figure 1a, trichoblast cells are split into two clusters. Why? Is one cluster from the inoculated sample and one from the mock-treated sample? Similarly, for the cortex clusters, do the different layers of cortex have different transcriptional profiles? This also relates to my comment above about the utility of additional UMAP visualizations.

As mentioned, the trichoblast cells have been split into two clusters. However, both clusters contain cells from both the control and treatment samples. This is also displayed in Supplemental Figure 3 (now Supplemental Figure 4). We have additionally added the figure below to show the integration of the two conditions. Although we can not be certain, due to the lack of known gene markers, it is likely that the separation of the two clusters is based on the developmental state of the cells.

An alternative cell type annotation approach is to first use cell type specific markers to annotate the mock-treated control sample. Then, the label transfer function in Seurat can be used to annotate the inoculated dataset. With this approach, it should be possible to identify cell types in the inoculated dataset that are absent in the mock-treated sample. Can the authors clarify why they chose to assign cell types based on the integrated inoculated and mock-treated data? This approach requires the assumption that inoculation does not substantially alter root cell identities. Do the nodule and bacteroid cells not constitute unique cell types?

We appreciate your suggestion and have tried this alternative approach. For this purpose, we annotated the control samples independently with the corresponding cell type. Next, we used the label transfer function in Seurat to annotate the treatment cells based on the control samples. Although this approach was effective in annotating the main cell types, it failed to identify the specific types of cells that respond to rhizobium inoculation. This may be because the cells involved in the infection and nodule development maintain their original cell identities as root hair and cortex cells as a consequence of the label transfer. The UMAP for the inoculated samples, colored by the predicted cell type using this approach, can be seen in the plot below.

Line 107: Do the proportions of the recovered cell types in the scRNA-seq data match the proportions of Lotus root cell types in vivo?

Based on the cross section shown in Figure 1 of Gavrilovic et al., 2016 and the longitudinal sections of Nadziejka et al., 2019, we roughly estimate the ratio of epidermis (including trichoblasts and atrichoblasts), cortex and endodermis cells as 1 : 2.25 : 0.5 (80, 90 and 20 cells per section and about 80, 40 and 40 μm long). In our 10dpi dataset, the ratio is about 1 : 1.75 : 0.2 (5200 cells : 9100 cells : 1100 cells). In our 5 dpi dataset the ratio is 1 : 1.72 : 0.3 (5000 cells : 8600 cells : 1500 cells). Both datasets more or less seem to reflect what we calculated from microscopy pictures. While we have recovered all expected cell types, it is likely that the proportion of each cell type will vary to different degrees with respect to the original proportions due to various factors associated with protoplasting. However, we think that the cells of interest in the symbiotic response have been identified in sufficient numbers to provide biologically meaningful data.

Line 109: Can the authors use a more quantitative metric to explain what is a ‘substantial’ transcriptional response to infection? In Figure 1C, neither the plot itself nor the legend indicate the cutoff for a gene to be considered differentially expressed (e.g., log fold change of 2?).

We selected the genes that have an adjusted p-value ≤ 0.05 and a log fold change > 0.25 . The cutoffs, previously only mentioned in the Methods, have been added to the main text.

Line 152, section entitled ‘Root hair and cortical infection transcriptional programs differ.’ The title of this section in the main text suggests that it is the unique programs that are noteworthy. However, the title of the Figure 3 legend (line 473; ‘Root hair- and cortex-IT cells share common gene sets’) highlights the opposite finding. Can the authors clarify if one of these results is of particular interest and why?

We have changed the title to “Infection responses in root hairs and cortical cells overlap but have distinct components” to provide a more balanced statement.

Line 171: Of the 32,180 high quality cells, please indicate in the main text how many are from the cyclops mutant and WT genotypes with mock treatment or inoculation, respectively.

We have indicated in the main text the number of cells from the WT and cyclops samples for the 5 dpi dataset.

The table quantifying the number of cells from each sample/cluster in the 5 dpi dataset can be found in Supplemental file 1 “5 dpi Cells_cluster”

Lines 242-247: I don't quite understand the big-picture significance of the symrkl1 phenotype. Is the mutant unable to fix nitrogen?

By demonstrating that a loss of SYMRKL1 leads to an infection phenotype, which does not affect nodule development or nitrogen fixation (there are no nitrogen-starvation symptoms at leaves or obvious growth impairment), we establish our approach as a powerful means of linking a very specific expression pattern to an associated phenotype. This is an advantage over classical genetic screens, which generally require severe phenotypes that are easy and fast to identify.

Line 435, Data availability: I encourage the authors to make the final Seurat objects available for download in addition to the sequencing data. This prevents the need for community members to re-analyze the raw data in order to explore it further. Although the scRNA-seq analysis-related methods are reasonably thorough, it is helpful to publish all code (e.g., via GitHub) and I encourage the authors to do this for reproducibility. I like the Shiny app as a resource for researchers who are interested in visualizing the final datasets. It is user-friendly and has a large amount of available information.

The Seurat objects have been uploaded on Figshare (<https://doi.org/10.6084/m9.figshare.23986200.v2>) and the scripts used for the data analyses are available on GitHub: https://github.com/LaviFechete/Lotus_Single_Cell

Minor comments

Line 101: The authors indicate that they carried out scRNA-seq of protoplasts from mock-treated and rhizobium-inoculated Lotus roots ‘in duplicates.’ Based on the methods (line 341), the duplicates are specifically biological replicates. Please clarify this detail in the main text. Further, can the authors clarify in the main text that 25,024 is the final number of cells after filtering out low quality cells.

We elaborated on the sample composition in lines 103 and 104 for the 10 dpi dataset and in lines 194-199 for the 5 dpi data.

Figure 1 b and c: The cluster numbers are very small and hard to see, especially when on top of the very small colored boxes. The dots in the dot plot are also very small and difficult to see.

We have increased the font size of the cluster numbers and the respective legend in Figure 1 and 4 and repositioned them in cases the cluster color might make it difficult to see the cluster number.

Figure 1c: In the legend, please indicate if the genes are up-regulated in the infected sample relative to the control or vice versa.

We have changed the description to “Down- and Upregulation of genes in the respective clusters in response to rhizobial infection compared to wild-type.”

Line 106 and Supplemental Figures 1 and 2: The figure legends for Supp Figures 1 and 2 indicate that scRNA-seq data are shown for both 10dpi and 5dpi samples. However, at the first mention of these supplemental datasets in the main text (line 101), only 10dpi samples have been described in detail. I assume that the 5dpi samples in these figures are from the WT controls run alongside the cyclops mutant. Please clarify this in the main text and figure legends.

We have added a brief description in lines 194 to 195.

Line 110: The least responsive tissues were phloem, QC, and xylem. Can the authors clarify if this finding was expected for xylem and phloem given their location deep in the center of the root?

The results are in line with previously reported responses (Cervantes-Pérez et al. 2022). While the location may be one explanation, it should be noted that other deep tissues such as pericycle are very responsive so we have not proposed such a link.

Line 519, Supp Fig 3: The cell type annotation legend for the cluster numbers and colors should be included as in Fig 1a.

Thank you for the suggestion. We have changed the figure accordingly.

Reviewer #3 (Remarks to the Author):

This is a very impressive single cell analysis of the transcriptome on the various cell types making up a *L. japonicus* (determinate) nodule. I found the work to be highly novel and a major piece of work for the field of nodulation. The identification of *symrkl1* from its expression in a cell specific manner and the absence of its expression in a *Cyclops* mutant is extremely impressive and novel. That *Symrkl1* is an important protein is clear from the aberrant infection threads formed by two independent *Lore1* mutants of *symrkl1*. It is clear

that traditional forward screens would not have identified the role of SYMRKL1 as the mutation does not alter an easily screened phenotype such as reduced nodule number or nitrogen starvation resulting from impaired N₂ fixation. My one suggestion to the investigators relates to the absence of a nodulation phenotype in the Lore1 mutants. As far as I can tell 0.5 ml of a 0.02-0.05 OD₆₀₀ culture of Mesorhizobium lotus R7A were used as an inoculum. As a rule of thumb most rhizobia have approx 10⁹ cells per ml at OD₆₀₀ 1, so 0.5 ml of a 0.02 OD₆₀₀ culture has approx 2 x 10⁷ cells and OD_{0.05} has approx 2.5 x 10⁷ bacterial cells. These are staggering high inocula and microbiologists working in competition assays would inoculate 10³- 10⁴ rhizobia on each plant. My point is if the nodulation experiments were done with 10³- 10⁴ R7A per plant there may well be a difference in final nodule number or in the timing of nodule formation (i.e. nodule appearance over time) between wildtype and symrkl1 mutants. This would show why the role of SYMRKL1 is critical in the real work of soil in the field with natural numbers of infecting rhizobia. Of course there may be no difference but it would be the cherry on top of what is already a great piece of work.

We thank the reviewer for this interesting suggestion and have performed the suggested experiments using R7A OD₆₀₀ values between 2E-7 to 2E-4. We have added the results to the manuscript as Supplemental Figure 12. We did not find a difference in nodule number 21 dpi.

REVIEWERS' COMMENTS

Reviewer #1 (Remarks to the Author):

Dear Authors,

Thank you for your thorough responses to my comments. After revision, I believe this paper is well-suited for publication at Nature Communications.

Reviewer #2 (Remarks to the Author):

I would really like to thank the authors for their clear and very thoughtful responses to my comments. All of my concerns have been addressed in the revised manuscript.

Response to reviewer's comments

Reviewer #1 (Remarks to the Author):

Dear Authors,

Thank you for your thorough responses to my comments. After revision, I believe this paper is well-suited for publication at Nature Communications.

Reviewer #2 (Remarks to the Author):

I would really like to thank the authors for their clear and very thoughtful responses to my comments. All of my concerns have been addressed in the revised manuscript.

Response:

We wish to thank all reviewers for their careful evaluation of our manuscript. We also highly appreciated their constructive comments, which helped us to significantly improve the quality of the manuscript.